# Sleep Time, Physical Activity, and Screen Time among Montana American Indian Youth

**DOI:** 10.3390/ijerph20176658

**Published:** 2023-08-26

**Authors:** Vernon Grant, Francine C. Gachupin

**Affiliations:** 1Center for American Indian and Rural Health Equity, Montana State University, Bozeman, MT 59718, USA; 2Department of Family and Community Medicine, College of Medicine, University of Arizona, Tucson, AZ 85716, USA; fcgachupin@arizona.edu

**Keywords:** tribal communities, health promotion, disease prevention, health disparity

## Abstract

The purpose of this study is to describe sleep, PA, and screen time behaviors among rural American Indian (AI) youth, stratified by sex and grade, to better understand how to address these health behaviors in AI youth. Body composition, a screen time survey, and demographic information were collected from 65 AI youth. Accelerometers were worn for 7 days. Sixty percent were overweight or obese. Sleep did not differ by sex or grade, with an actigraphy-based total sleep time (aTST) of 7.8 h per night. Boys had significantly more light PA (*p* = 0.002) and vigorous PA (*p* = 0.01) compared to girls. Screen time did differ by sex but not by grade, with girls in the sixth and seventh grades reporting more screen time than boys, but boys in the eighth grade reporting more screen time than girls. Despite sex differences in screen time, high levels of screen time and obesity and low levels of PA and sleep are a concern in this population.

## 1. Introduction

Sleep, physical activity (PA), and screen time are well-known behaviors associated with obesity in youth [1,2,3]. These behaviors are different for boys and girls [1,3]. Although high rates of obesity have been reported among American Indian (AI) youth [4,5], relatively little is known about sixth–eighth graders (11 to 13 year olds).

Sleep is essential for youth because it plays an important role in their physical and mental development. Most teens should get between 8 and 10 h of sleep per night [6,7]. Unfortunately, research indicates that many young people get far less sleep than they need. Sex differences suggest that 53.1% of boys and 58.5% of girls get insufficient sleep, with an average sleep duration of 7.2 h [8]. A study assessing longitudinal changes in sleep from grades 7 to 9 found that both boys and girls get significantly less sleep as they age (girls: 8.4 h to 7.8 h; boys: 8.6 h to 7.9 h) [9]. Another study found that predictors of short sleep time were being female, being non-white, and being in higher grades [10]. Girls report a higher prevalence of waking up tired, difficulty falling asleep, lower sleep quality, and waking up in the middle of the night than boys [7,11,12]. Accordingly, the first aim of our study is to understand sleep time in AI youth.

PA is perhaps one of the most beneficial health behaviors [13]. Regular PA can help youth improve their cardiorespiratory fitness, build strong bones and muscles, control their weight, reduce symptoms of anxiety and depression, and reduce the risk of developing health conditions such as obesity, diabetes, heart disease, cancer, and high blood pressure [13]. Research shows that PA declines as youth age [14]. Studies have reported that between 38% and 65% of youth, aged 9–15 years old, meet the recommended 60 min of PA per day [15,16]. These studies further report that between 41.4% and 81.2% of boys and between 22.7% and 60.1% of girls meet the recommended 60 min of PA per day [17,18]. Research consistently shows that boys engage in more PA minutes per day than girls [14,15,18,19,20,21,22,23]. Studies have also reported that youth in rural areas spend less time in moderate-to-vigorous physical activity (MVPA) compared to youth in urban areas [15]. The second aim of our study is to understand PA in rural AI youth, and any differences between sexes.

With an increase in the use of and access to digital technology, there has been an increase in average screen time exposure for youth [24,25]. A study reported that the average screen time in youth in grades 4–6 was 3.4 h and increased to 4.6 h by the 9th and 10th grades [26]. According to the National Health and Examination Survey (NHANES) data, approximately 65% of youth engage in high amounts of screen time, which is defined as ≥2 h per day [19]. Other studies show that 20.1% of youth engage in at least 3 h of screen time per day [16] and another reported that 51% of youth engaged in >5 h of daily screen time [27]. Poulain et al. [28] reported that 100% of 850 German young people, aged 10–17 years old, averaged 5.3 h of daily screen time. In terms of Indigenous youth, a meta-analysis by Foulds et al. [29] revealed that Indigenous boys engage in more screen time compared to Indigenous girls (3.3 h vs. 2.7 h). Other studies have reported similar findings, concluding that boys consistently have higher amounts of screen time compared to girls [20,22,30].

By the sixth grade, approximately 80% of young people own a cellular telephone [31]. Girls who own cell phones report compulsive texting, excessive social media use, and a strong need to stay in contact with their peers [9,32]. In contrast, boys spend more time playing video games [33]. The third aim of our study is to understand screen time in rural AI youth, especially in reference to changes with age. A study has shown that youth engage in less screen time as they age, except for those whose parents have a low education level [34].

These health behaviors are not siloed and are associated with each other. For example, the amount of screen time youth engage in is directly linked to both sleep and PA. Upwards of 97% of young people have at least one media device in their bedroom, which has been associated with later bedtimes, less time in bed, increased sleep onset latency, later wake times, and shorter sleep durations [35]. Studies show that the longer the amount of time youth spend engaged in television viewing and online social networking, the more sleep problems exist [36]. In terms of PA, increasing age and being female are related to a higher odds for low PA (defined as engaging in active play six times or less per week) and high amounts of screen time (defined as more than 2 h per day) [19]. Additionally, low PA is associated with high amounts of screen time [19]. Accordingly, this work will test the associations between screen time, actigraphy-based total sleep time (aTST), and PA.

AI youth have the highest prevalence of obesity of all the ethnic groups in the United States (USA) [37], placing them at disproportionate risk for adult obesity and obesity-driven metabolic diseases, and clearly, sleep, PA, and screen time are contributing behaviors. The environmental influence of living in high-poverty areas that are characteristic of reservation communities also impact behavior and health. For instance, social determinants of health have been shown to be a significant factor in the development of obesity and chronic disease [38,39,40]. Another significant factor impacting behavior and disease risk is historical trauma. Historical trauma has roots in colonization and the subsequent events associated with it. Colonialization has impacted generations of AI people and brought about a host of issues, not least of which are racism, disease, and suicide [41,42,43]. Behavioral interventions, particularly when adopted in early life, can instill healthy behaviors and circumvent a trajectory of life-long chronic disease. The first step, then, is to understand these behaviors among AI youth.

The purpose of this study is to describe sleep, physical activity, and screen time behaviors among rural youth, stratified by sex and grade. Because of the exploratory nature of this work, we did not test any specific hypotheses.

## 2. Materials and Methods

### 2.1. Study Design

This cross-sectional study assessed sleep, PA, screen time, and demographic variables in 6th–8th-grade youth in a Montana middle school located in a tribal community.

### 2.2. Procedure

Our research team collaborated with the school district and middle school principals to conduct this study. Participants were recruited at parent–teacher conferences where a booth was set up and parents could sign a parental consent form after the study purpose and activities were described. The investigator also introduced the study to every student when they attended scheduled Physical Education (PE) classes. Every student is required to complete PE every day, which is why we selected this particular class to recruit students. Interested students were given a parental consent form to take home and students who returned a signed parental consent form were enrolled in the study. There were no exclusion criteria—all students enrolled in school were eligible to participate.

The study was conducted in Spring 2017 in two separate buildings (the 6th graders were in a separate building from 7th and 8th graders). In each building, the principal provided classroom space and allowed the research team to collect data during lunch to minimize disruption to classroom instruction. The students completed a child assent form before participating. All instructions were explained to each participant, and if they needed help, a research team member read the assent form and survey questions to the participant(s). The students were given a demographic and screen time survey, and their height and weight were measured. Before they left, the students were fit with an Actical accelerometer (Philips Respironics, Bend, OR, USA) secured on their non-dominant wrist by a waterproof wristband to wear for seven days. If they removed it, they were asked to return it to the office. The accelerometers were collected the following week, and each participant received a nominal monetary incentive.

### 2.3. Demographics

The demographic information collected included age, grade, and sex. Height was measured using a portable stadiometer to the nearest 0.1 cm (Seca Model 217, Seca, Inc., Hanover, MD, USA). An electronic scale was used to measure weight (Tanita Model BWB-800S, Tanita, Inc., Chicago, IL, USA) without shoes to the nearest 0.1 kg. Height and weight were converted to age- and sex-specific BMI percentiles according to the Centers for Disease Control algorithms [44].

### 2.4. Actigraphy: Physical Activity and Sleep

PA and sleep were collected by the Actical wrist accelerometer for 7 consecutive days. The epochs were set for 1 min. Actical discriminates among different PA intensities (e.g., sedentary, light, moderate, and vigorous) using built-in cut-off points, which were used to calculate the mean overall PA for sex and grade. The following sleep measures were calculated by hand: actigraphy-based time in bed (aTIB), defined as the time from the first minute of uninterrupted lying down in the evening (“in bed”) to the last minute scored as lying down the next morning (“out of bed”); aTST, computed as the sum of all epochs scored as sedentary during aTIB; and sleep efficiency, calculated as aTST divided by aTIB [45].

### 2.5. Screen Time

Screen time was assessed using a seven-item survey developed by the Healthy Children, Strong Families research team, who also assessed screen time [46]. Four items focused on the total minutes spent watching television, total minutes spent using the computer, total minutes spent playing video games, and overall screen time (all screen time variables combined) were used for this study.

### 2.6. Data Analysis

Actical software was used to download the wrist accelerometer data, which were then exported to an Excel spreadsheet. The 7-day raw actigraphy data were then transferred to R 3.3.0 (R Foundation for Statistical Computing, Vienna, Austria) to format the PA and sleep variables. A total of 58 (89.2%) Actical wrist accelerometers contained usable data. Unusable data was attributed to participants cutting off the device, resulting in excessive non-wear time. An MVPA category was created by summing moderate and vigorous PA data. The mean value for overall activity was calculated for each participant.

Overall screen time (minutes) was determined by summing the total minutes spent watching television, total minutes spent using the computer, and total minutes spent playing video games.

All demographic, screen time, PA, and sleep data were analyzed in R. The assumptions of linear regression were fitted to the data in separate models using aTST, MVPA, and screen time as dependent variables. Each unadjusted model assessed whether the dependent variable between genders was dependent on grade. Because it was unknown how the *p*-value would be impacted after adjusting for other variables in the model, adjusted models were computed despite an insignificant p-value in the unadjusted models. ANOVA was used to fit the data.

## 3. Results

### 3.1. Demographics

A total of 65 young people participated in this study (Table 1). There were 24 (36.9%) sixth graders, 25 (38.4%) seventh graders, and 16 (24.6%) eighth graders; there were 28 (43.1%) boys and 37 (56.9%) girls. The body composition results classified one (1.5%) student as underweight, 25 (38.4%) students as normal weight, and 39 (59.9%) students as overweight or obese.

### 3.2. Sleep Time, Physical Activity, and Screen Time

The mean and standard deviation for the PA, aTST, and screen time variables are provided in Table 1. Boys had significantly more light physical activity (*p* = 0.002) and vigorous physical activity (0.01) compared to girls. There were no significant differences between boys and girls for the sleep and screen time variables.

### 3.3. Unadjusted and Adjusted aTST, MVPA, and Screen Time Models

An unadjusted and adjusted models assessing the association of aTST, MVPA, and screen time between boys and girls by grade is contained in Table 2. According to both models, there is no evidence that the difference in mean aTST between boys and girls depends on grade level after accounting for age, BMI percentile, MVPA, and screen time (*p*-value = 0.512, F-stat = 0.679, F-distribution on 2 and 43 df). There is no evidence that the difference in MVPA between boys and girls depends on grade level after accounting for age, BMI percentile, total sleep time, and screen time (*p*-value = 0.153, F-stat = 1.961, F-distribution on 2 and 43 df). The unadjusted model shows weak evidence that screen time between boys and girls is dependent upon grade (*p*-value = 0.07479, F-stat = 2.742, F-distribution on 2 and 47 df). The adjusted model suggests that screen time between boys and girls is dependent upon grade after accounting for BMI percentile, age, aTST, and MVPA (*p*-value = 0.03465, F-stat = 3.64, F-distribution on 2 and 43 df). No other significant associations were found for the MVPA and aTST adjusted models.

## 4. Discussion

The purpose of this study is to determine whether aTST, MVPA, and screen time between boys and girls depends on grade level in an under-studied and relatively unknown population—youth attending middle school in an isolated AI tribal community. We found that aTST and MVPA between boys and girls do not depend on grade level. However, we did find that screen time between boys and girls is dependent upon grade level. One of the notable findings from this work is that 39 (60%) young people in this sample were either overweight or obese. The data show that girls obtain more aTST than boys in the sixth and seventh grades, but boys obtain more aTST than girls in the eighth grade. Boys engage in more MVPA compared to girls for all grades. Finally, girls engage in more screen time than boys in the sixth and seventh grades, but boys engage in more screen time than girls in the eighth grade.

There was no evidence that the difference in mean aTST between boys and girls depends on grade level. aTST revealed that girls sleep more than boys in the sixth and seventh grades, whereas boys sleep more than girls in the eighth grade. Studies report that youth between the ages of 6 and 16 have delayed bedtimes, shortened time in bed, and variable wake times with each additional year of age [47]. This suggests that youth are developing lower sleep quality and quantity as they age. The highest amount of aTST was recorded for seventh-grade girls, at 483.2 min (approximately 8 h). Despite these estimates, aTST averaged around 462.8 min (approximately 7.7 h), with eighth-grade girls having the lowest amount of aTST, at a value below the sleep recommendations for this age group [48]. It appears that AI youth have similar sleep estimates to those in studies that include non-Indian populations. In the general population, boys trend towards higher levels of sleep duration compared to girls as they age [10]. In a study of approximately 7349 young people in grades 9–12 in Ontario and Alberta, Canada, girls were more likely to be classified in the short sleep duration group compared to boys [8]. Other work states that girls tend to have lower sleep quality than boys [7,11,12]. For instance, de Matos et al. [11] report that girls have a greater tendency to wake up tired, have difficulty falling asleep, and wake up in the middle of the night compared to boys. The prevalence of insomnia symptoms such as difficulty initiating sleep, difficulty maintaining sleep, and early morning awakenings significantly increases as youth report higher puberty scores, with girls reporting a higher rate of insomnia symptoms compared to boys [49]. Significant differences between boys and girls are observed in preference for morning or evening at the age of 16 years, suggesting that boys are more evening-oriented than girls, but girls show shifts in preference for evening around adolescence, and then, toward morning around the age of 21 years [50]. Accordingly, youth in the puberty stage report later bed times [51]. Girls tend to go through puberty earlier than boys, which may explain the shift in aTST between girls and boys from grades 7 to 8, where girls tend to sleep more in the earlier grades and boys tend to sleep more in the later grades. Understanding sleep and how to address sleep issues in youth is important for public health as adequate sleep is linked to lower levels of anxiety, depression, irritability, delinquency, and aggression [6]. This is especially true for AI youth living on the reservation. Objective sleep measures are sparse in this population and gaining a deeper understanding of this important health behavior will help researchers to effectively address this issue. However, more work is needed to understand the sleep environment and family sleep routines in AI youth living on the reservation.

Our analysis revealed weak evidence that the difference in MVPA between genders depends on grade level. In this reservation community, boys engage in more MVPA than girls in all grades (sixth–eighth). Nader et al. [14] assessed 1032 young people aged 9–15 years and found that age and gender were the most important determinants of MVPA. However, this study also found that low family income, lower BMI percentile, and location of residence also significantly impacted MVPA, which were not assessed in our models. Another variable that has been shown to be a significant factor in engaging in MVPA in this age group is sports participation [52], which was also not assessed. Data collection was carried out in late spring towards the end of school. Organized sports were not available at that time, which may have had an impact on MVPA levels in each gender. Our work is consistent with prior research that has shown that girls consistently engage in significantly less MVPA compared to boys [14,15,18,19,21] and are much more sedentary as they reach pubertal age [33]. Studies show that as girls age and reach puberty, they engage in less MVPA [16,23], which may explain why girls were less active in this age group. PA can be complex to assess, as studies have shown that there are significant weekday-to-weekend differences [15,17,18] and time of day is an important factor for its assessment (e.g., before school, during the day, after school, and in the evening) [17,18]. Thus, time of day may be more important to understand than the average MVPA score throughout the day. This may provide a better indication of how to assess PA in this population. Another aspect of AI youth living on the reservation is that youth in rural areas are much less likely to achieve PA recommendations compared to youth who live in urban areas [15]. Perhaps we did not observe an effect in the adjusted model because of the rural nature of the reservation and lack of opportunity for PA. Low family income has been shown to be an important determinant of youth engaging in MVPA [14] and reflects the socioeconomic circumstances of this sample [53]. The data collection time (late spring), rural setting, and lack of PA opportunities may have all contributed to the sensitivity of the model to help inform the hypothesis that MVPA between genders is grade-dependent. Additionally, this work provides insight into the PA levels of a population that is unknown.

Our model shows that screen time between genders is dependent upon grade level. The data show that girls engage in more screen time compared to boys in the sixth and seventh grades, whereas boys engage in more screen time compared to girls in the eighth grade. Aligned with these findings, we also found that approximately 60% (39) of young people in this sample were overweight or obese. Studies show that obese youth engage in high levels of screen time (>2 h per day) [16] and obese youth are consistently shown to engage in higher amounts of screen time and do not adhere to the recommended screen time levels [16]. A study reported that overweight Indigenous youth watch television more than normal-weight indigenous youth [29]. This study also found that Indigenous overweight youth watch significantly more television than normal-weight youth (2.72 h per day compared to 2.34 h per day) [29]. According to NHANES data, approximately 75% of boys and 66% of girls aged 9–11 years report high screen time (>2 h per day) [19]. A closer look at the data reveals that factors that most increase the odds of engaging in 2 or more hours per day of screen time are older age and being female [19]. Other work suggests that boys engage in higher levels of screen time compared to girls [22,30], and as youth age and enter higher grades, they are more likely to engage in higher levels of screen time [33]. We observed similar screen time trends in this sample of AI youth. Work by Marques et al. [23] show that girls post-puberty have significantly higher amounts of screen time compared to girls pre-puberty. Perhaps screen selection is an important factor in this behavior. A recent paper reported that the factor that most increases the odds of youth aged 10–14 years spending more than 2 h per day in cell phone screen time was gender, specifically, being female [52]. Upwards of 80% of youth own a cell phone by the time they reach the sixth grade [31]. Our analysis did not assess cell phone screen time. We combined television, computer, and video games to compute an overall screen time estimate. However, cell phone use may explain screen time more effectively than the types of screen time we measured. A study of 8th- and 11th-grade youth found that girls have the urge to stay in contact with their peers and engage in significantly more compulsive texting compared to boys [32]. Additionally, girls spend more time on social media compared to boys [9]. It appears that screen time plays a significant role in the amount of MVPA and aTST youth engage in. This is true for AI youth living in this tribal community.

### Strengths and Limitations

This study was cross-sectional and included a small sample with volunteer participants. School start time [10] and sleep hygiene [7] have significant impacts on youth sleep time, neither of which was measured in this study. It is important to note that actigraphy-based aTST cannot determine sleep quality and cannot detect sleep disorders such as sleep apnea. The use of aTST, assessed via accelerometry, simply provides a quantitative estimate of sleep. Another important variable that impacts sleep, PA, and screen time is puberty. The literature appears to weigh puberty as a significant factor in this age group [23,49]. Prior research shows that as youth reach the pubertal stages in life, girls experience difficulty falling asleep [47]. Youth of pubertal age report greater difficulties with sleep, such as later bed times and insomnia-related symptoms [49,51]. Puberty was not measured and should be considered when working with youth. The strengths of this study include our use of a study population of AI youth living in a tribal community that have never been reported in the literature. Prior to this work, sleep time, PA, and screen time were unknown. Sleep and PA were measured via accelerometry, which enhances the strength of these estimates. We found that aTST and MVPA between genders do not depend upon grade level, but screen time does. The implications of these findings suggest that a deeper understanding of screen time should be developed, specifically, assessing screen time preference in terms of cell phones or video games and whether sex has an influence on a certain type of screen time. This will provide researchers with a better understanding of how to address the epidemic of youth spending most of their waking hours consumed with screen time. Accordingly, future work measuring screen time should assess cell phone and social media use. Sleep measures should assess sleep quality, sleep disorders, and sleep hygiene. Future work should also assess puberty in order to better understand how these confounding variables (screen time, sleep, and puberty) contribute to unhealthy behaviors. Environmental and historical factors such as social determinants of health and historic trauma may provide an in-depth understanding of barriers to health. The outcome of this work may reveal other factors not assessed and may provide future directions for interventions designed specifically for AI tribal youth to limit their screen time and enhance their sleep and PA behaviors. Tribal communities are characteristically rural with limited PA opportunities outside of structured sports. Additionally, we successfully collected data from 65 participants, which is a significant proportion of the population in this small reservation community.

## 5. Conclusions

Limited data are available on sleep, PA, and screen time behaviors in AI youth, and no studies have been conducted in this isolated tribal community. We assessed objective measures of sleep, PA, screen time, and body composition in 65 young people in the sixth–eighth grades living in a tribal community. Univariate associations between sex show that boys had significantly more light physical activity (*p* = 0.002) and vigorous physical activity (0.01) compared to girls. However, there were no significant differences between sex for sleep and screen time. When the adjusted models were assessed, we found that aTST and MVPA between genders do not depend upon grade level, but screen time does. This suggests that girls are not getting adequate PA, and depending on what grade students are in, screen time is a major concern. That 60% (n = 39) of the sample was either overweight or obese further underscores that the lack of PA and prolonged daily screen time should be addressed in this population.

## Figures and Tables

**Table 1 ijerph-20-06658-t001:** Mean and standard deviation of age, MVPA *, aTST **, screen time, and BMI % ^&^ for AI boys and girls (n = 65).

	Boys (n = 8)	Girls (n = 16)		Boys (n = 13)	Girls (n = 12)		Boys (n = 7)	Girls (n = 9)
6th Grade			7th Grade			8th Grade		
Age (years)	14.5 (0.5)	14.3 (0.5)	Age (years)	15.6 (0.6)	15.7 (0.4)	Age (years)	16.5 (0.5)	16.6 (0.5)
MVPA * (minutes)	224.2 (164.3)	172.8 (50.6)	MVPA * (minutes)	181.6 (77.4)	141.8 (38.0)	MVPA * (minutes)	214.2 (67.5)	174.7 (74.1)
aTST ** (minutes)	460.8 (108.1)	466.5 (34.1)	aTST ** (minutes)	482.5 (34.7)	483.2 (51.4)	aTST ** (minutes)	466.3 (31.1)	459.9 (81.3)
Screen Time (minutes)	110 (83.1)	118.7 (82.3)	Screen Time (minutes)	112.6 (86.9)	170.4 (219.8)	Screen Time (minutes)	267.8 (295.0)	125 (159.8)
BMI % ^&^			BMI % ^&^			BMI % ^&^		
Underweight	1 (12.5%)	0 (0%)	Underweight	0 (0%)	0 (0%)	Underweight	0 (%)	0 (0%)
Normal Weight	3 (37.5%)	6 (37.5%)	Normal Weight	4 (30.7%)	4 (33.3%)	Normal Weight	3 (42.8%)	5 (55.5%)
Overweight	0 (0%)	6 (37.5%)	Overweight	3 (23.0%)	5 (41.6%)	Overweight	1 (14.2%)	0 (0%)
Obese	4 (50%)	4 (25%)	Obese	6 (46.1%)	3 (25%)	Obese	3 (42.8%)	4 (44.4%)

* MVPA = moderate-to-vigorous physical activity (min); ** aTST = total sleep time (min); ^&^ BMI % = BMI percentile.

**Table 2 ijerph-20-06658-t002:** Unadjusted and adjusted models for MVPA *, aTST **, and screen time.

	Unadjusted aTST * (min) Model		Adjusted aTST * (min) Model			Unadjusted MVPA ** (min) Model		Adjusted MVPA ** (min) Model			Unadjusted Screen Time (min) Model		Adjusted Screen Time (min) Model	
	β	*p*-Value	β	*p*-Value		β	*p*-Value	β	*p*-Value		β	*p*-Value	β	*p*-Value
BMI % ^&^			298.6	0.71	BMI % ^&^			364.2	0.76	BMI % ^&^			69,297	0.09
Age			506.1	0.65	Age			1239	0.58	Age			24,852	0.30
MVPA ** (min)			1338	0.47	aTST * (min)			2405	0.44	aTST * (min)			6313	0.60
Screen Time (min)			1141	0.50	Screen Time (min)			13115	0.07	MVPA ** (min)			82,516	0.06
Gender	222.4	0.76	222.4	0.77	Gender	70.3	0.90	70.3	0.89	Gender	9581	0.54	9581	0.52
Grade	4958	0.35	4958	0.38	Grade	359.3	0.96	359.3	0.95	Grade	20,274	0.67	20,274	0.64
Gender × Grade	4220	0.41	3357	0.51	Gender × Grade	9120	0.34	15,547	0.15	Gender × Grade	141,374	0.07	169,418	0.03

* aTST = total sleep time; ** MVPA = moderate-to-vigorous physical activity; ^&^ BMI % = body mass index percentile.

## Data Availability

The data presented in this manuscript were obtained through an NIH-funded research collaboration with a tribe in Montana (identity confidential). They were generated through the actions of the research team and in partnership with the tribe. The community has approved the publication and dissemination of the data in the format provided in this manuscript. The primary dataset from our study is owned by the tribe, in recognition of tribal sovereign authority stipulated by the government of the United States. Requests for access to the primary data can be made to the tribal leadership. While we cannot speak for them, we believe that all requests would be given due consideration. Such requests can be directed to: Vernon Grant, Center for American Indian and Rural Health Equity, Montana State University, 2155 Analysis Drive, Bozeman, MT 59718.

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
