# Peer review of "Sleep Time, Physical Activity, and Screen Time among Montana American Indian Youth"

_ijerph, 2023, doi:10.3390/ijerph20176658_

Round 1
Reviewer 1 Report
Dear Authors,
the conducted research is not groundbreaking to science, but it is well done in terms of methodology.
Comments:
1. Keywords: Why was "obesity" entered? For research purposes, there is no information about body composition/nutrition tests, etc.
2. Introduction: I sugest to standardize "4.6 hours" and "5.31" (lines 55 and 61). Either hundredths or decimals.
3. Introduction: no hypotheses mentioned in the Discussion (line 193).
4. Conclusion: the first sentence is not a conclusion. The last sentence is not the conclusion of your research. It's more suited to Discussion or Strengths adn Limitations. Please refer only to your research in your conclusions. Detailed information, not generalities unrelated to your research. Please keep the relationship: research objective - conclusions. The mention of obesity and being overweight lends itself to Discussion. Your research goal was not to evaluate certain phenomena in the context of obesity or overweight.
Reviewer 2 Report
This article reports on sleep, physical activity (PA) and screen time patterns among Montana American Indian youth. Authors contend that while high rates of obesity have been reported among American Indian (AI) youth, much less is known about 6th – 8th graders. Furthermore, high levels of screen time and obesity and low levels of PA and sleep are a concern in this population.
The review is as follows.
1. Lines 82-85 – In “AI youth have the highest prevalence of obesity of all ethnic groups in the United 82 States (U.S.)[37], placing them at disproportionate risk for adult obesity and obesity-driven metabolic diseases and clearly sleep, PA, and screen time are contributing behaviors”, it will be important for the authors to contextualize this information for the reader and comment of the structural, sociocultural, historical and other factors that contribute to the high prevalence of obesity among AI youth. For instance, a mention of the social determinants of health would be germane to this discussion.
2. Regarding study recruitment, what was the response rate? Were there inclusion and exclusion criteria?
3. How long was the survey? Were incentives given for participation?
4. In the discussion, authors should expand on the implications of this research as well as potential future directions of this research.
Overall, this is a relevant and insightful paper. It is on a unique topic that could add to the existing literature. There should be expanded discussion on the social determinants of health of sleep, physical activity and screen time patterns among Montana American Indian youth. There should be additional discussion on study recruitment within the Materials and Met
Reviewer 3 Report
The manuscript titled “Sleep, physical activity, and screen time patterns among Montana American Indian Youth” compares sleep, physical activity (PA) and screen-time cross-sectionally in 65 young adult youth stratified by sex and grade. A descriptive understanding of the influence of sex and age (grade) differences in sleep, PA and screen time behaviors is important for improving American Indian health (obesity) through targeted programs. However, the stated purpose of the paper to “better understand the associations (of sleep, PA and screen time) with obesity has not been realized in the current manuscript. Significance of the findings would be greatly improved if this analysis was performed and reported.
Major Concerns
· The abstract states “The purpose of this study is to describe sleep, PA, and screen time behaviors among rural youth, stratified by sex and grade, to better understand associations with obesity”, however how the authors intend to investigate this (aims and hypotheses ) are not clearly defined in the introduction.
Specifically, the rationale, objectives and analytic approach to understand the relationship between all three health behaviors and obesity has not been assessed.
· The rationale for the models reported in Table 2 are not set out in the aim and hypotheses.
· The authors calculation of sleep measures by hand is understandably the best approach given the data format available. However, obtaining an average actigraphy-based total sleep time (aTST) might not be particularly useful metric (especially given the null findings).
o There are considerable limitations in using actigraphy as a proxy for sleep, especially in the context of obesity as it does not adequately assess the impact on sleep quality from sleep disorders such as sleep apnea. These limitations must be addressed in the manuscript.
Minor Comments
· Please use sleep “time” instead of “pattern”. The only measure of sleep employed in the study is aTST. Reporting sleep patterns per se requires more detailed metrics of sleep.
· Use an “a” or some other notation in front of the sleep measures (i.e. aTIB, aTST) to indicate the measure is actigraphy-based rather than via the gold standard polysomnography (PSG).
· Please indicate the significant differences in the outcomes according to sex and grade stratifications in Table 1
Reviewer 4 Report
Dear Authors,
this is a very interesting topic, especially that high percentage of youth is obese or overweight. Based on the results of the study, campaigns should be introduced to encourage young people to engage in regular physical activity.
I have got a few questions:
Line 17
„higher light PA” more people had light PA or higher level of PA?
Material and methods:
“The investigator also introduced the study to every student 98 when they attended scheduled Physical Education class” what about students that do not attend PE classes for any reason?
A study was conducted in 2017 it is quite a long time ago.
Line 118’without shoes’. Tanita is a body composition analyzer that determines body weight through electrodes and their contact through bare feet or feet and hands. Please describe the procedure of mastering body mass.
How was the PA measured? By heart rate?
Line 174 “There is weak evidence…” – p=0.153 – there is no evidence
Round 2
Reviewer 2 Report
The authors clearly address the requested feedback and had a good mention of social determinants of health and historical trauma. There is also good discussion on implications for future research. The revised paper is clearer and improved.